# Choosing between the UN's alternative views of population aging

**Warren C. Sanderson**[1,2,3]*, **Sergei Scherbov**[1,4]

**1** Wittgenstein Centre for Demography and Global Human Capital (Univ. Vienna, IIASA, VID/ÖAW), International Institute for Applied Systems Analysis, Laxenburg, Austria, **2** Department of Economics, Stony Brook University, Stony Brook, New York, United States of America, **3** CUNY Institute for Demographic Research, New York, New York, United States of America, **4** International Laboratory on Demography and Human Capital, Russian Presidential Academy of National Economy and Public Administration, Prospekt Vernadskogo, Moscow, Russian Federation

* warren.sanderson@stonybrook.edu.

**Data Availability Statement:** The data underlying the results presented in the study are available from University of California, Berkeley, Max Planck Institute for Demographic Research. Human Mortality Database [Internet]. 2019 [cited 2019 Jun

## Abstract

Commonly used measures of population aging categorize adults into those who are "old" and those who are not. How this threshold of the stage of "old age" is determined is crucial for our understanding of population aging. We propose that the old age threshold be determined using an equivalency criterion. People at the old age threshold should be roughly equivalent to one another in relevant characteristics regardless of when and where they lived. The UN publishes two variants of the potential support ratio based on different old age thresholds. One old age threshold is based on a fixed chronological age and the other on a fixed remaining life expectancy. Using historical data on 5-year death rates at the old age threshold as an indicator of one aspect of health, we assess the extent to which the two approaches are consistent with the equivalency criterion. The death rates are derived from all the complete cohort life tables in the Human Mortality Database. We show that the old age threshold based on a fixed remaining life expectancy is consistent with the equivalency criterion, while the old age threshold based on a fixed chronological age is not. The picture of population aging that emerges when measures consistent with the equivalency criterion are used are markedly different from those that result when the equivalency criterion is violated. We recommend that measures of aging that violate the equivalency criterion should only be used in special circumstances where that violation is unimportant.

## Introduction

One widely used measure of population aging is the potential support ratio, the inverse of the old age dependency ratio [1]. The potential support ratio divides the population 20 years of age and older into two disjoint age groups. Conceptually, the ratio is meant to reflect the stages of the human life cycle, distinguishing between adults who are elderly and those who are not. To compute the ratio two sorts of information are needed, the number of people at each age starting from 20 and a threshold age that divides the adult population into a group who are elderly and a group who are not.

30]. Available from: http://www.mortality.org
Profiles of Ageing 2019 [Internet]. [cited 2019 Jul 24]. Available from: https://population.un.org/ProfilesOfAgeing2019/index.html

**Funding:** This project has received funding from the European Union's Horizon 2020 research and innovation program under grant agreement No 635316 (Project Name: Ageing Trajectories of Health: Longitudinal Opportunities and Synergies, ATHLOS), SS. The funders had no role in study design, data collection and analysis, decision to publish, or preparation of the manuscript. The International Institute for Applied Systems Analysis encourages and actively supports its researchers to publish their research in journal articles or books that are made available for free to all users (gold open access).

**Competing interests:** The authors have declared that no competing interests exist.

On its website *Profiles of Ageing 2019* [1] the UN now publishes a conventional potential support ratio (PSR) and a prospective potential support ratio (PPSR). The difference between the two variants is based solely on different threshold ages at which people first become categorized as"old" [1]. In the PSR that threshold age is age 65 and is fixed independent of time or place. In the PPSR the threshold age is the age where remaining life expectancy is 15 years. We call the first, the *conventional old age threshold (COAT)* and the second the *prospective old age threshold (POAT)*. The *COAT* is the most commonly used old age threshold, but it has the disadvantage that it does not change over time and is the same for all countries regardless of their trajectories of aging.

The choice of whether to use the *COAT* or the *POAT* in assessing the extent of population aging is not arbitrary. It is not like choosing between Celsius and Fahrenheit in the measurement of temperature. Having measures of population aging based on the *COAT* and the *POAT* is like having two kinds of thermometers, where sometimes both indicate that the temperature is increasing and sometimes one indicates it is getting warmer while the other indicates it is getting cooler. Indeed, sometimes measures of population aging based on the two old age thresholds change in the same direction and sometimes they do not.

The purpose of this article is to evaluate the two old age thresholds based on an equivalency criterion. People at the old age threshold should be similar to one another in dimensions that are relevant to the study of population aging. Here we test the equivalency criterion using 5-year death rates, an indicator of health that is comparable across time and space. If an old age threshold is consistent with the equivalency criterion, then people at the old age threshold would be similar to one another in terms of 5-year death rates regardless of when or where they lived. If the equivalency criterion is violated then people in one country with a certain 5-year death rate would be categorized as old, while people in another country with the same 5-year death rate would not be.

We proceed in three steps. First, we show that the choice of which old age threshold to use is consequential. We do this by showing that the potential support ratios that use the *COAT* and the *POAT* produce very different histories and potential future paths of population aging. In the second step, we evaluate the *COAT* and the *POAT* using the equivalency criterion. We find that the *POAT* is consistent with the equivalency criterion, while the *COAT* is not. We conclude with a discussion of the implications of using measures of population aging that are consistent or not consistent with the equivalency criterion.

## Two views of population aging

In this paper, we evaluate whether the two old age thresholds used in the UN's *Profiles of Ageing 2019* (1) are consistent with the equivalency criterion. We do this because measures of population aging using the two thresholds produce different views of the history and likely future of population aging. To illustrate those differences, we use potential support ratios computed with each of the thresholds. An example of the differences in the two prospective support ratios is presented in Fig 1, where we present the conventional potential support ratio (PSR) and the prospective potential support ratio (PPSR) for the world from 1950 to 2050 as they appear in the UN's *Profiles of Ageing 2019* (1). The figures for 1950 to 2015 are estimates, while subsequent figures are forecasts. The PSR is based on the *COAT*. It is the ratio of people 20–64 years-old to those 65+ years-old. Thus, the PSR assumes that people should be classified as being old beginning at age 65 regardless of where they live and when they lived. The PPSR is based on the *POAT*, where the age at which people are classified as old varies over time and space. More information about the *POAT* can be found in [2–5].

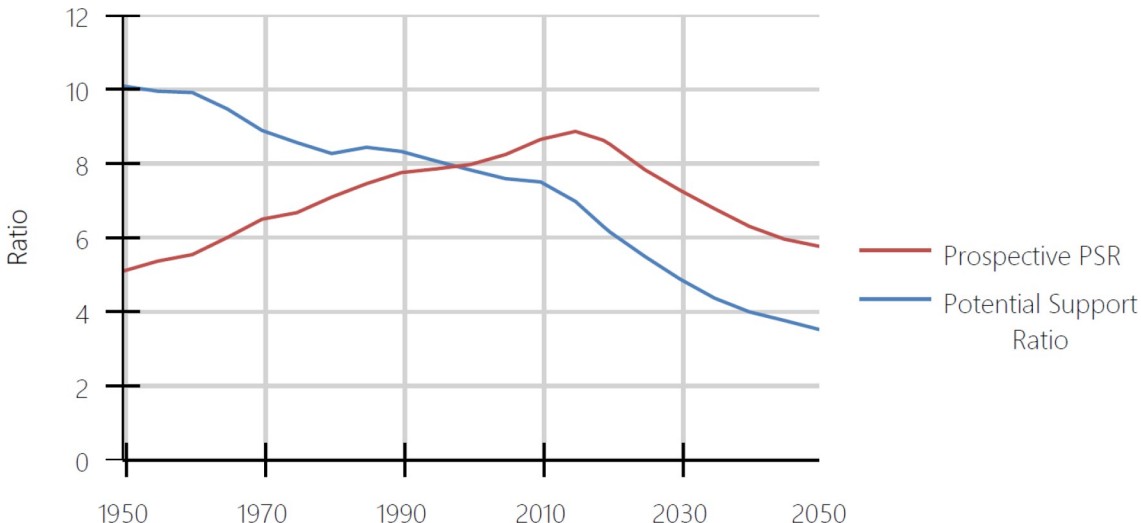

**Fig 1. Potential support ratio and prospective potential support ratio, World, 1950–2050.** Source: United Nations, Profiles of Ageing 2019 [1].

Both the PSR and the PPSR measure the aging of populations. In particular, an increasing PSR or PPSR indicates that the population is growing younger in the sense that for each elderly person in the population there are more younger adults.

In Fig 1, the PSR falls continuously over the period, indicating that the population of the world was getting older. The decline in the PSR was slower between 1950 and 2010 and more rapid after that. In contrast, the PPSR tells a different story. The PPSR rises between 1950 and 2015. During this period, the PPSR shows that the world's population was getting younger and only begins to age after 2015. In 2050, the PPSR indicates that the world's population will be about aged as it was in 1965. The use of different old age thresholds leads to different conclusions about whether the world's population only began to get older after 2015 or whether it was aging continuously from 1950 onward.

We present a second example of the implications of using each of the two old age thresholds in Fig 2, where we compare the behavior of the PSR and the PPSR in China. The Chinese economy achieved a rate of economic growth of around 10 percent per annum from the economic reforms at the end of the 1970s to the early part of the current decade. [6] According to the PSR, this high and sustained rate of economic growth occurred during a period in which the population of China was rapidly aging. The PSR fell from around 10 in 1978 to around 7 in 2013. In contrast, during the same period, the PPSR rose from around 7 in 1978 to around 8 in 2013. Did China's period of rapid economic growth occur while its population was aging relatively rapidly or slowly becoming younger? Understanding the relationship between population aging and economic growth in China depends on which old age threshold is used.

Having two different answers to questions like whether the world's population was aging between 1950 and 2015 and whether China's most rapid period of economic growth occurred during a period of population aging or one where the population was growing younger is a problem. The problem arises because the PSR and the PPSR measure population aging differently. The difference arises only because of differences in their threshold ages, the ages which separate older adults from younger ones. The assessment of population aging depends crucially on which threshold age is assumed.

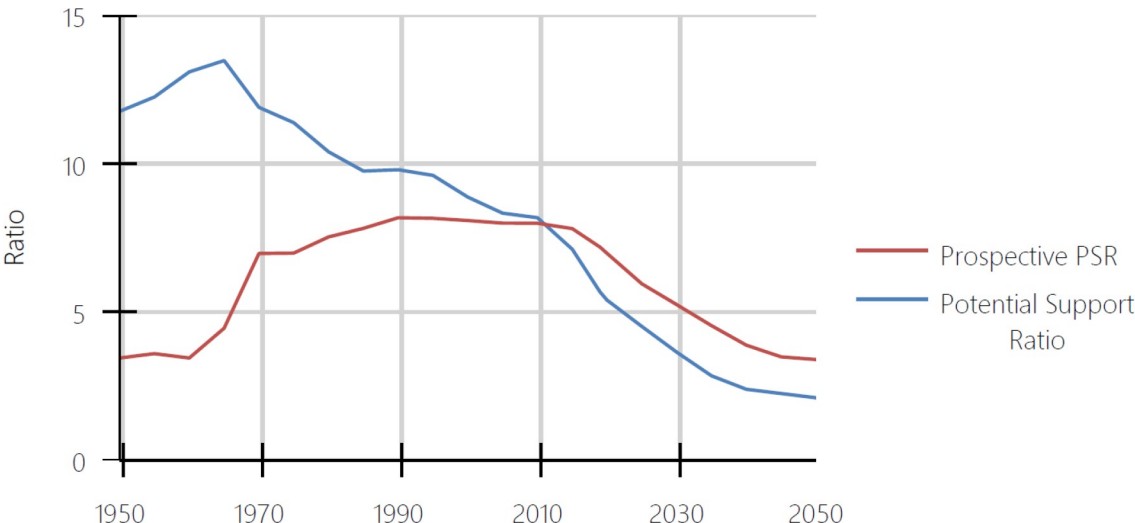

**Fig 2. Potential support ratio and prospective potential support ratio, China, 1950–2050.** Source: United Nations, Profiles of Ageing 2019 [1].

## Threshold ages and the equivalency criterion

Threshold ages are meant to reflect a fundamental aspect of human life. If people are fortunate enough, they eventually grow old. However, what it means to grow old in different places and at different times is certainly different [7,8]. Threshold ages are not literally meant to distinguish between people who are supported by others from those who are not. They are meant to reflect, on a population level, our understanding that old age is a distinct life cycle stage. Threshold ages, then, are a tool for translating our intuitive ideas about old age into population level measures.

We evaluate different old age thresholds based on an equivalency criterion. People at the old age thresholds in different countries and in different years should be similar to one another in ways that are relevant for the understanding of population aging. Here, we focus on the similarity of people at the old age threshold in terms of their 5-year death rates. The advantage of using 5-year death rates is that they are consistently and reliably measured for many populations over long periods. An old age threshold would be consistent with the equivalency criterion if people at the old age threshold in different countries or at different times would have about the same 5-year death rate. An old age threshold would be inconsistent with the equivalency criterion if people at the old age threshold in different countries or at different times had very different 5-year death rates. In that case, people with the same 5-year death rates would be counted as old in one context and not old in another.

## Evaluating the two old age thresholds

At this writing, the Human Mortality Database [9] has complete cohort life tables for Denmark, England and Wales, Finland, France, Iceland, Italy, the Netherlands, Norway, Scotland, Sweden, and Switzerland. The longest time series of single year of birth cohort life tables comes from Sweden where the life tables cover people born from 1751 to 1927. The shortest time series come from Finland (birth cohorts from 1878 to 1927) and Switzerland (birth cohorts from 1876 to 1925). In Fig 3, we present the two threshold ages computed from these cohort life tables.

In Fig 3, the conventional threshold age is a horizontal line at age 65. Although the figure shows data for eleven locations and many decades, only one conventional threshold age is

## Prospective Old−Age Threshold

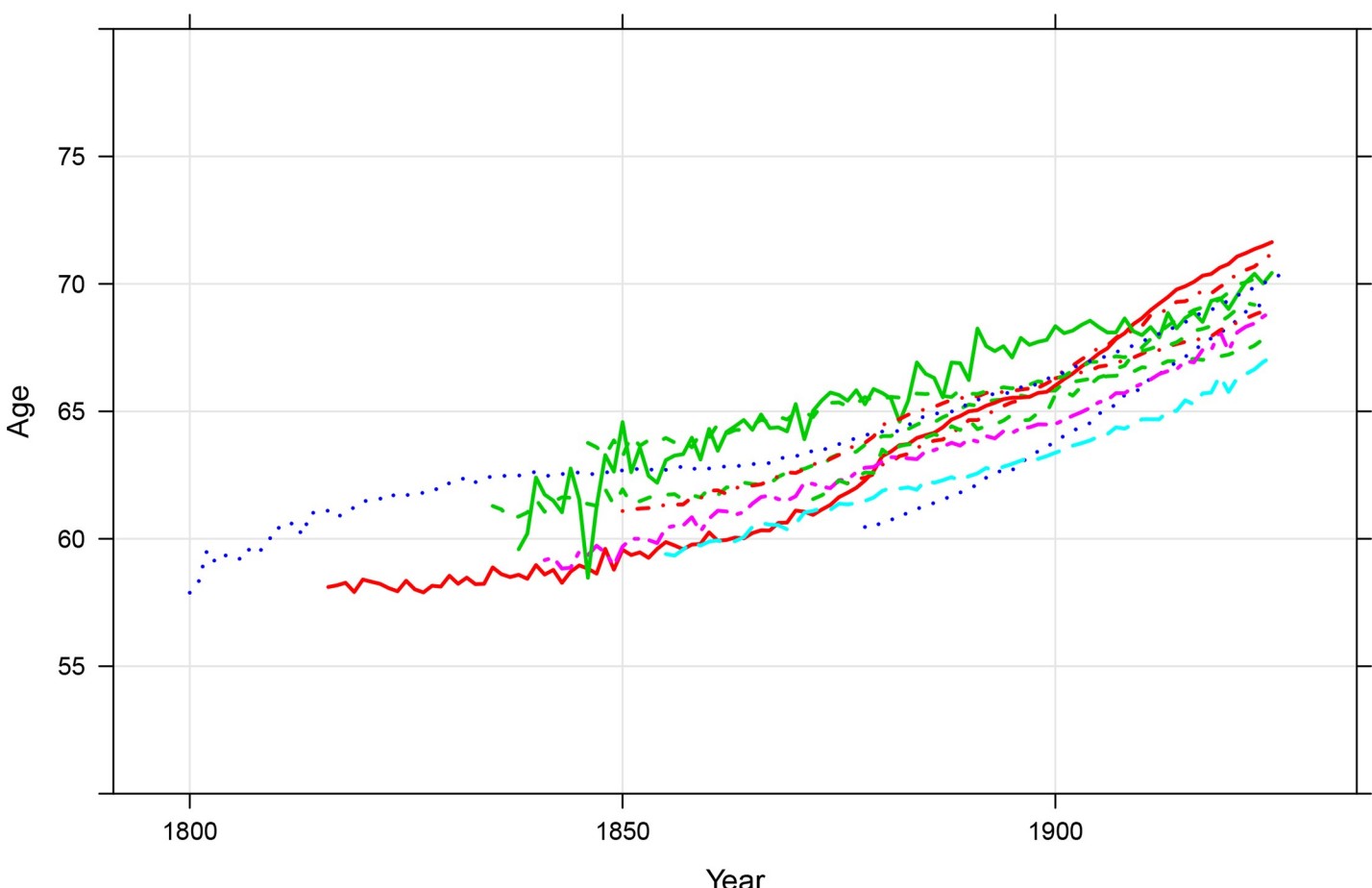

**Fig 3. Prospective old age threshold (age at which remaining life expectancy is 15 years), from 1830 to latest observation, cohort life tables from Denmark, England and Wales, Finland, France, Iceland, Italy, the Netherlands, Norway, Scotland, Sweden, and Switzerland, both sexes combined.** Source: Human Mortality Database [9] and Authors' Calculations. Underlying data are presented in the Supplementary Information.

drawn because that age is fixed and does not change over time or space. The prospective threshold ages differ both across locations and over time. For example, the prospective threshold age for the Swedish cohort born in 1830 was around four years higher than the prospective threshold age for the French cohort born in the same year. The speed of increase in the prospective threshold ages also differed. For example, the French cohort born in 1920 had a prospective threshold age that was around one year older the Swedish cohort born in that year.

We evaluate the two threshold ages using 5-year death rates. The equivalency criterion says that people at the old age threshold should be similar to one another in some ways relevant to the understanding of population aging regardless of where and when they lived. Fig 4 shows the 5-year death rate at the *POAT* and Fig 5 shows the death rate at the *COAT*. The 5-year death rate is a reflection of one aspect of health [10,11]. There are potentially other relevant characteristics, but the 5-year death rate at specific ages has the advantage that it has been consistently calculated over a long period for many countries. In Fig 4, we see that the prospective old age threshold is consistent with the equivalency criterion and we see from Fig 5 that the conventional one is not. The approximate constancy of 5-year death rates at the prospective

## 5−year Death Rate at POAT

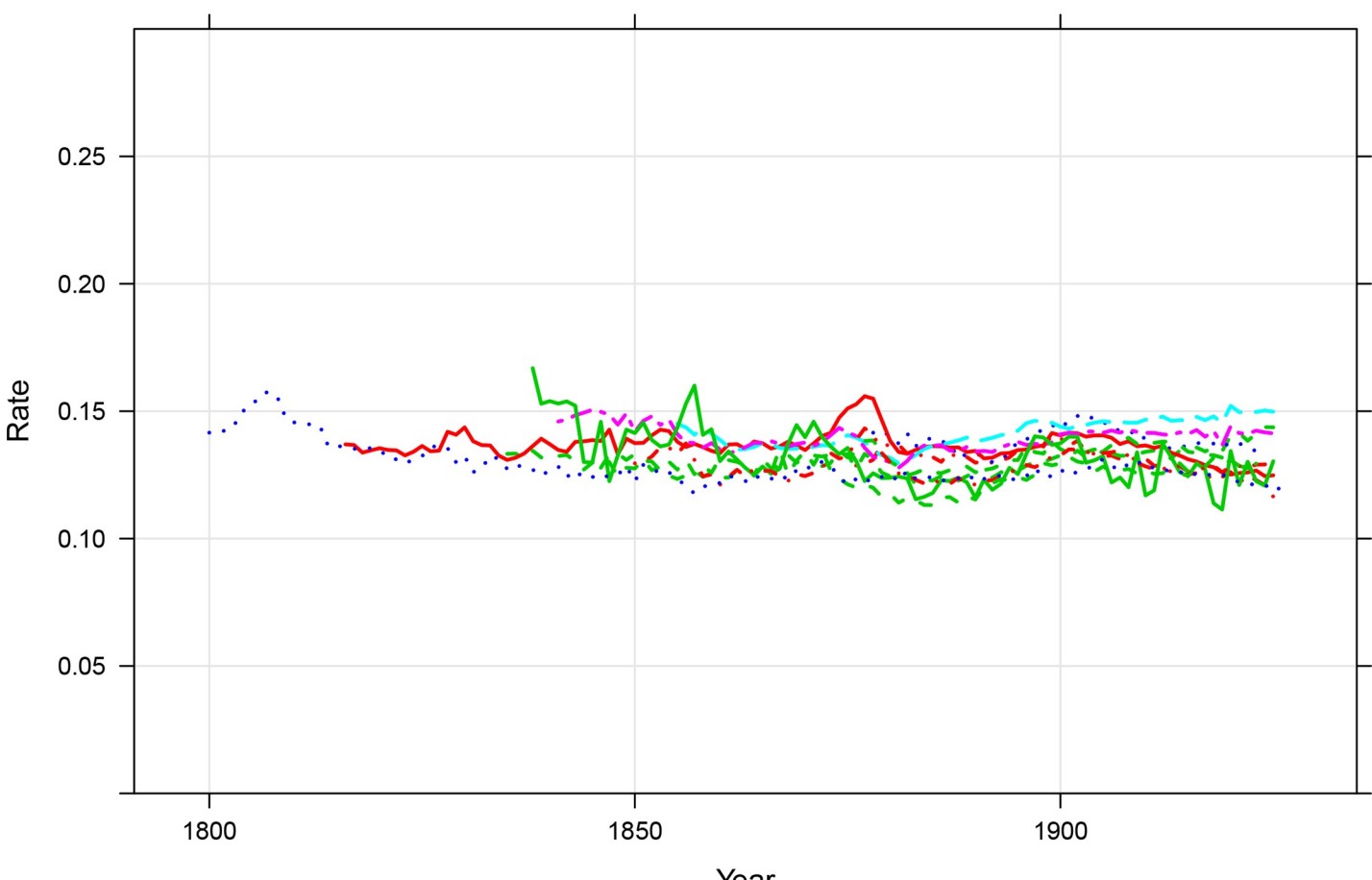

**Fig 4. 5-Year death rates at the prospective old age threshold, 1830 to the latest observation, cohort life tables from Denmark, England and Wales, Finland, France, Iceland, Italy, the Netherlands, Norway, Scotland, Sweden, and Switzerland, both sexes combined.** Source: Human Mortality Database [9] and Authors' Calculations. Underlying data are presented in the Supplementary Information.

old age threshold is particularly noteworthy because of the variations in the prospective thresholds shown in Fig 3.

## Two views of population aging revisited

In Fig 1, we presented a graph taken from the UN's *Profiles of Ageing 2019*. The figure showed that the world's population was growing older from 1950 to 2015 when assessed using the conventional potential support ratio and was growing younger when assessed using the prospective one. The difference from those two views of our history stems from when people become categorized as old. The UN provides two old age thresholds, the *COAT* and the *POAT*. In the previous section, we showed that the *COAT* was not consistent with the equivalency criterion. People at the *COAT* differed significantly from one another in terms of the 5-year death rates depending on when and where they lived. We also showed that the *POAT* was consistent with the equivalency criterion. People at the *POAT* were similar to one another in terms of 5-year death rates across countries and decades.

## 5−year Death Rate at Age 65

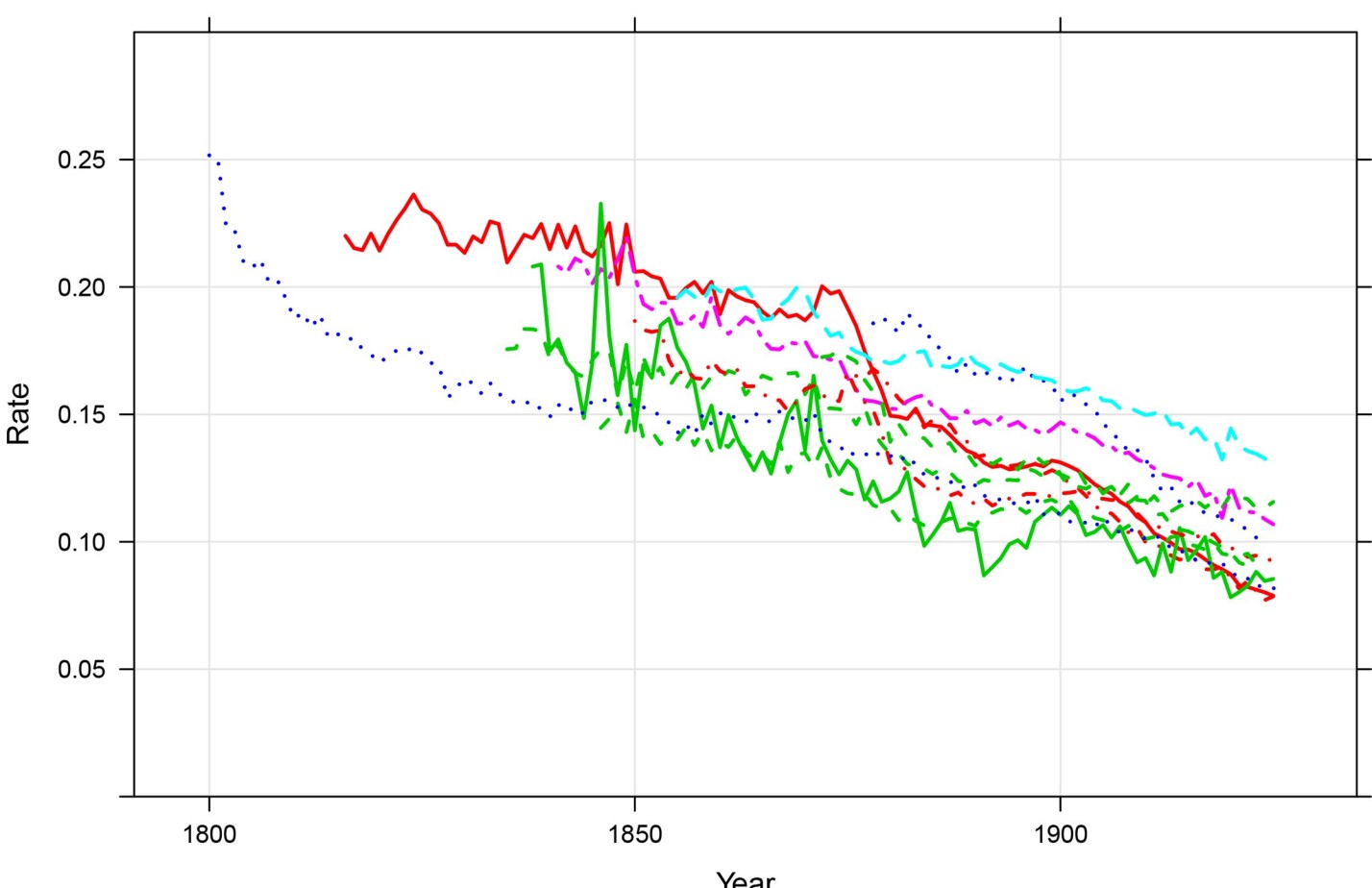

**Fig 5. 5-Year death rates at age 65, 1830 to the latest observation, cohort life tables from Denmark, England and Wales, Finland, France, Iceland, Italy, the Netherlands, Norway, Scotland, Sweden, and Switzerland, both sexes combined.** Source: Human Mortality Database [9] and Authors' Calculations. Underlying data are presented in the Supplementary Information.

The equivalency criterion defines sets of characteristic-equivalent ages. In doing so, it also has implications for the entire group of people who are at or above those characteristic-equivalent ages. For example, when the *POAT* is used, the group of people who are categorized as potentially needing support are people who are at ages where remaining life expectancy is 15 years or less. Those same people are also at the age where 5-year death rates are greater than around 0.14. When the *COAT* is used the groups of people who are categorized as potentially needing support are neither consistent in terms of remaining life expectancy nor 5-year death rates. They are only consistent in people having had their 65th birthdays.

The UN's *Profiles of Ageing 2019* provides people who study population aging with a choice of perspectives. There, whether the world's population grew older or younger from 1950 to 2015cannot be answered without deciding first on how to assess who is elderly. We recommend the use of the equivalency criterion in making that decision because it defines who is elderly in a consistent way based on characteristics relevant to the study of population aging.

## Supporting information

**S1 File.**
(XLSM)

**S2 File.**
(XLSX)

## Author Contributions

**Conceptualization:** Warren C. Sanderson, Sergei Scherbov.

**Formal analysis:** Warren C. Sanderson, Sergei Scherbov.

**Methodology:** Warren C. Sanderson, Sergei Scherbov.

**Visualization:** Warren C. Sanderson, Sergei Scherbov.

**Writing – original draft:** Warren C. Sanderson, Sergei Scherbov.

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
