## [Decision Letter · Decision Letter 0]

26 Nov 2019

PONE-D-19-23563

Choosing Between the UN’s Alternative Views of Population Aging

PLOS ONE

Dear Dr. Sanderson,

Thank you for submitting your manuscript to PLOS ONE. After careful consideration, we feel that it has merit but does not fully meet PLOS ONE’s publication criteria as it currently stands. Therefore, we invite you to submit a revised version of the manuscript that addresses the points raised during the review process.

The reviews for your manuscript were quite mixed: 1 accept with revisions and 1 reject. Given the timeliness of the topic, I think this paper could make an important contribution, but I should be upfront by saying that if you do revise and resubmit your paper, it may be potentially sent to a new reviewer.

We would appreciate receiving your revised manuscript by Jan 10 2020 11:59PM. To enhance the reproducibility of your results, we recommend that if applicable you deposit your laboratory protocols in protocols.io, where a protocol can be assigned its own identifier (DOI) such that it can be cited independently in the future. For instructions see: http://journals.plos.org/plosone/s/submission-guidelines#loc-laboratory-protocols

We look forward to receiving your revised manuscript.

Kind regards,

Alison Gemmill, PhD, MPH

Academic Editor

PLOS ONE

Journal Requirements:

a) Please provide an amended Funding Statement that declares *all* the funding or sources of support received during this specific study (whether external or internal to your organization) as detailed online in our guide for authors at http://journals.plos.org/plosone/s/submit-now.  

b) Please state what role the funders took in the study.  If any authors received a salary from any of your funders, please state which authors and which funder. If the funders had no role, please state: "The funders had no role in study design, data collection and analysis, decision to publish, or preparation of the manuscript."

Please include your amended statements

Reviewers' comments:

Reviewer's Responses to Questions

**Comments to the Author**

1. Is the manuscript technically sound, and do the data support the conclusions?

Reviewer #1: Partly

Reviewer #2: Yes

2. Has the statistical analysis been performed appropriately and rigorously? 

Reviewer #1: N/A

Reviewer #2: Yes

3. Have the authors made all data underlying the findings in their manuscript fully available?

Reviewer #1: No

Reviewer #2: Yes

4. Is the manuscript presented in an intelligible fashion and written in standard English?

Reviewer #1: No

Reviewer #2: Yes

5. Review Comments to the Author

Reviewer #1: The topic of this manuscript is relevant and important. However, the manuscript shows many shortcomings and requires a substantial revision before being published. The main issue is that it lacks innovation. The authors summarize correctly the implications of the 2 UN variants to compute the potential support ratio, but essentially do not go beyond this point. In other words, the manuscript lacks innovation. And every reader with some basic demographic and algebra knowledge is able to understand the implications of the 2 variants for population aging without this summary, but just by looking at the UN webpage and the formulas there. So, what is the innovation of this paper? What new aspects do we learn from this summary?

An aspect that can be further developed is the conceptualization and evaluation of the different threshold ages. This has potential, but again, it’s not well presented and enough to convince a reader.

Reviewer #2: Dear authors and editors,

General remarks:

Thanks for this nice clear article. It's an easy read, it sets a clear criterion, and presents convincing evidence for homegeneity in the case of age thresholds benchmarked to a remaining life expectancy of 15. The text could be revised to remove some redundancies (for example l183-191 is repetition of concepts already stated). I make one suggestion that would increase the number of series in your figures 3 and 4, and this is coupled with a suggested redux of fig 4. It would be easy for an experienced researcher to replicate this study from the same sources, but it is preferable and IMO best practice if you make the exercises of Figs 3 and 4 reproducible in the strict sense: tidy up your code and stick it in a repository such as OSF, and share a link in the article. In any case, I think a conditional accept is in order. The conditions are a copy edit, code repo, and consideration of the below suggestion.

Best regards,

Tim Riffe

Main suggestion:

fig 4: Why did you choose these six countries? I count 11 HMD populations with cohort lifetables. Not that HMD cohort lifetables are restricted to cohorts observed from birth until approximate extinction. Both series can stretch into earlier cohorts than those shown, because the HMD also publishes raw cohort death rates (cMx) also for incomplete cohorts. They could also extend to the right if you were willing to use some humble extrapolation to close out not-yet extinct cohorts. I suggest looking at the MortalityLaws R package to make this easy if you go this route. How far to extrapolate to this closeout? Maybe only so far as it doesn't make a huge difference which closeout law you pick? Extending left (no extrapolation required) and extending right (if you have the gumption) would increase the count of usable countries beyond the 11 with long series of complete cohort lifetables (or the six you chose). In that case I'd invite you to use all series of adequate length and to rethink the design of this figure, as it's an effective piece of evidence, but could be even more effective. Why not just two panels, one with age 65 5-year death rates for all countries and the other with all the 5-year death rate around POAT? There will be no point in trying to make each series distinguishable, so I suggest you make them all the same desaturated color, with only a couple series highlighted (and directly labelled) that you actually talk about in the text. This ought to work because the trends within each color in the present fig 4 are mostly parallel and that's the point. It may end up being a stronger case. A similar principle could be applied in Fig 3 (more countries, and only highlight the ones your refer to).

The flat red lines of fig 4 make me wonder if the same would hold for death rates around other life-expectancy benchmarked threshold ages, like e(x*) = [5, 10, 15, 20, 25]. The UN doesn't use that, though, so maybe this is beside the point? Your call.

Small things:

First paragraph l21-27: Consider removing second sentence (l22-23) and moving the present 3rd sentence (l23-25) to the end of the paragraph. i.e. how to calculate followed by what it's supposed to represent. Would be nice to include the most common threshold age here already to have a clear idea.

l31 space before citaition (2)

l58-59 ", assumes that the age at which people are classified as elderly should vary over time and space" : The PPSR threshold age, in contrast, is sensitive to mortality conditions, and so the indicator is also sensitive to both the population structure and mortality conditions in each time and place.

l113-114 you can strike this sentence I think.

l138 year -> years

6. PLOS authors have the option to publish the peer review history of their article (what does this mean?). If published, this will include your full peer review and any attached files.

Reviewer #1: No

Reviewer #2: Yes: Tim Riffe

---

## [Author Response · Author response to Decision Letter 0]

30 Mar 2020

Responses to Reviewers’ Comments:

Choosing Between the UN’s Alternative Views of Population Aging

Authors: Warren C. Sanderson, Sergei Scherbov

Funding: This project has received funding from the European Union’s Horizon 2020 research and innovation program under grant agreement No 635316 (Project Name: Ageing Trajectories of Health: Longitudinal Opportunities and Synergies, ATHLOS), SS. The funders had no role in study design, data collection and analysis, decision to publish, or preparation of the manuscript.

Please note that the Abstract and the Introduction have been extensively rewritten. The Abstract is entirely new. The original abstract can be found in the file that contains the original submission.

Reviewer 1: The manuscript lacks innovation

The manuscript has been revised to emphasize its innovation. Its innovation is the explicit use of an equivalency criterion in defining groups of older adults. The vast majority of the literature of population aging defines the group of older adults as starting at age 65. When this is done the resulting groups of older adults are grossly inconsistent with one another with respect characteristics important for the study of population aging, such as remaining life expectancy and health (defined using death rates). The manuscript previously did not make this point strongly enough. The current version of the manuscript uses the term “equivalency criterion” to make clear that this is not a manuscript just about two different measures of population aging. It is about the more general point that measures of population aging should define groups of older adults that are defined consistently with respect to important aspects of aging. 

The point of our paper is that definitions of the group of older adults should not be made arbitrarily but should be made with some criteria of equivalency in mind. Maintaining that the group of older adults be defined in a consistent and meaningful way is, in our view, an important contribution. We show in the paper that the use of a consistent and meaningful definition of the group of older adults changes materially our view of the history and likely future of population aging.

Reviewer 2: “In any case, I think a conditional accept is in order. The conditions are a copy edit, code repo, and consideration of the below suggestion.”

1. The paper has been copy edited and focused on its innovative contribution that groups of older adults be defined consistently with respect to characteristics relevant to the study of population aging.

2. All the data used in this article come from open access sources. Figures 1 and 2 are graphs published by the United Nations. Figures 3 and 4 use data from the Human Mortality Database. 

3. Reviewer 2’s suggestion had three parts. First, he suggested that we use more time series. He noted that the Human Mortality Database has 11 sets of cohort life tables. He suggested that we use all of them and we have done this now. Second, he suggested that Figures 3 and 4 be revised to use all 11 sets of cohort life tables. Third, he suggested that we extend incomplete cohort life tables and add them to the 11 that we used. This is an interesting suggestion, but we do not think that this is the right place to do it. If we were to do it, a great deal of the paper would have to be devoted to the technique of extending the cohort life tables and a sensitivity analysis of the results. This would distract from our main point that the definition of the group of older adults cannot be arbitrary but must be made on the basis of some relevant criteria. We look forward to the possibility of making use of the suggestion in a future paper.

Reviewer 2: Remove redundancies, line 183-189:

The text of lines 183-191 in the original manuscript read:

“Two threshold ages are used in the UN’s Profiles in Ageing 2019. Threshold ages should categorize the elderly into as homogeneous groups as possible on the basis of some relevant characteristics. Different threshold ages are appropriate when different characteristics of the elderly are studied. The prospective threshold age produces groups of elderly that are comparatively homogeneous with respect to five-year death rates and remaining life expectancy at the age when people first become categorized as elderly. The conventional threshold age produces groups that are homogenous with respect to a fixed age, in the sense that all people classified as elderly are at or above that chronological age. The choice to threshold age depends on which characteristics of populations are being studied.”

This text has been omitted from the manuscript.

Reviewer 2: revise paragraph including lines 22-25.

This has been done. We cannot give figures for threshold age here because the prospective old-age threshold has not yet been defined. We include figures for the prospective old-age threshold later in the manuscript.

Reviewer 2: space before reference 2 on line 31.

This has been corrected.

Reviewer 2: Suggested revision in the text at lines 58-59

l58-59 ", assumes that the age at which people are classified as elderly should vary over time and space" : The PPSR threshold age, in contrast, is sensitive to mortality conditions, and so the indicator is also sensitive to both the population structure and mortality conditions in each time and place.

This has been done. The text has been revised to make it similar to the reviewer’s suggestion.

Reviewer 2: Omit line 113-114.

This has been done.

Reviewer 2: change “year” to “years” in line 138

This has been done.

---

## [Decision Letter · Decision Letter 1]

1 May 2020

PONE-D-19-23563R1

Choosing Between the UN’s Alternative Views of Population Aging

PLOS ONE

Dear Dr. Sanderson,

Thank you for submitting your manuscript to PLOS ONE. After careful consideration, we feel that it has merit but does not fully meet PLOS ONE’s publication criteria as it currently stands. Therefore, we invite you to submit a revised version of the manuscript that addresses the points raised during the review process.

We would appreciate receiving your revised manuscript by Jun 15 2020 11:59PM. To enhance the reproducibility of your results, we recommend that if applicable you deposit your laboratory protocols in protocols.io, where a protocol can be assigned its own identifier (DOI) such that it can be cited independently in the future. For instructions see: http://journals.plos.org/plosone/s/submission-guidelines#loc-laboratory-protocols

One of the reviewers had minor comments in the revised manuscript. Please make sure to address these:

We look forward to receiving your revised manuscript.

Kind regards,

Alison Gemmill, PhD, MPH

Academic Editor

PLOS ONE

Reviewers' comments:

Reviewer's Responses to Questions

**Comments to the Author**

1. If the authors have adequately addressed your comments raised in a previous round of review and you feel that this manuscript is now acceptable for publication, you may indicate that here to bypass the “Comments to the Author” section, enter your conflict of interest statement in the “Confidential to Editor” section, and submit your "Accept" recommendation.

Reviewer #2: All comments have been addressed

2. Is the manuscript technically sound, and do the data support the conclusions?

Reviewer #2: Yes

3. Has the statistical analysis been performed appropriately and rigorously? 

Reviewer #2: Yes

4. Have the authors made all data underlying the findings in their manuscript fully available?

Reviewer #2: Yes

5. Is the manuscript presented in an intelligible fashion and written in standard English?

Reviewer #2: Yes

6. Review Comments to the Author

Reviewer #2: I saw a couple typos e.g line 166., authors should go through the whole thing.

Small thing: ordinate axis limit could be .3 for both fig 4 and 5

7. PLOS authors have the option to publish the peer review history of their article (what does this mean?). If published, this will include your full peer review and any attached files.

Reviewer #2: Yes: Tim Riffe

---

## [Author Response · Author response to Decision Letter 1]

7 May 2020

Reviewer #2: I saw a couple typos e.g line 166., authors should go through the whole thing.

This has been done.

Small thing: ordinate axis limit could be .3 for both fig 4 and 5

This has been done.

---

## [Editor Report · Decision Letter 2]

11 May 2020

Choosing Between the UN’s Alternative Views of Population Aging

PONE-D-19-23563R2

Dear Dr. Sanderson,

We are pleased to inform you that your manuscript has been judged scientifically suitable for publication and will be formally accepted for publication once it complies with all outstanding technical requirements.

With kind regards,

Alison Gemmill, PhD, MPH

Academic Editor

PLOS ONE
---

## [Editor Report · Acceptance letter]

1 Jun 2020

PONE-D-19-23563R2 

Choosing Between the UN’s Alternative Views of Population Aging 

Dear Dr. Sanderson:

I am pleased to inform you that your manuscript has been deemed suitable for publication in PLOS ONE. Congratulations! Your manuscript is now with our production department. 

With kind regards,

on behalf of

Dr. Alison Gemmill 

Academic Editor

PLOS ONE